# Urban Expansion Simulation Coupled with Residential Location Selection and Land Acquisition Bargaining: A Case Study of Wuhan Urban Development Zone, Central China's Hubei Province

**Heng Liu, Lu Zhou and Diwei Tang ***

College of Forestry and Horticulture, Hubei Minzu University, Enshi 445000, China
* Correspondence: jstdw@163.com

**Abstract:** The urban expansion process involves multiple stakeholders whose interactions and decision-making behaviors have a complex impact on urban land conversion. In this study, we established an urban expansion simulation model that couples two sub-models: the residential location selection model and the land acquisition bargaining model. Those sub-models include four types of agents: resident agent (RA), real estate developer agent (DA), government agent (GA), and farmer agent (FA). The residential location selection model is composed of three agents, RA, DA, and GA, and is first used to select residential locations, while an artificial neural network (ANN) is used to define the behavior rules of RA and RA selects pixels as candidate locations according to the joint decision probability. Then the land acquisition bargaining model is used, which is composed of GA and FA. If the land acquisition is successful, a pixel is converted into urban land, which is occupied by the corresponding RA; otherwise, the RA selects the next pixel and enters the bargaining process again, and so on, until the RA successfully selects a residential location. Each iteration represents the selection process of an agent. We used this model to simulate urban expansion within the Wuhan Urban Development Zone (WHUDZ) of central China from 2009 to 2019. The overall accuracy and Kappa coefficient of the simulation results were 92.78% and 55.24%, respectively, which were higher than the results using logistic regression cellular automata. Moreover, we obtained the relative contributions of various influencing factors in the ANN on the residential location selection, revealing the influence of the land acquisition process on land expansion. In addition, the coupled model predicted that the WHUDZ's urban land area will reach 1415.82 km$^2$ in 2029, mainly through extensional expansion, and the southeast and northwest will be expansion hot spots.

**Keywords:** urban expansion simulation; agent-based model; residential location selection; land acquisition bargaining; Wuhan Urban Development Zone

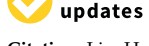


## 1. Introduction

In 2018, 55.3% of the world's population lived in cities, which is expected to increase to 60.0% by 2030 [1], indicating that urbanization will continue to advance globally in the coming decades. As the largest developing country in the world, China's urbanization level has increased from 17.9% in 1978 to 64.7% in 2021. However, during rapid urbanization, land tends to expand in a more "aggressive" way that is faster than population growth [2,3], and the characteristics of land-centered urbanization appear [4]. On the one hand, local governments have gained considerable fiscal revenue through land transfer, which has promoted regional economic development. On the other hand, the large-scale expropriation of farmers' land for urban construction has caused social and ecological problems, such as land acquisition conflicts and encroachment on extensive farmland. China's land resources have long faced the dual tasks of development and conservation, the contradictions between supply and demand, and the pressures of resource utilization and ecological protection [5].

Urban expansion simulation technology can guide urban expansion by predicting future land demand and simulating development scenarios, which can promote rational urban construction and efficient land use. Because it can provide decision support for land management and urban planning, it has become a research hotspot in geography and urban planning. Urban expansion simulation models can be divided into top-down and bottom-up models [6]. The top-down model mainly uses mathematical statistics or empirical equations to express the dynamic time changes and ignores the spatial–temporal differences of urban expansion, which fails to reflect the complexity of the urban system evolution process, such as the system dynamics model [7].

Bottom-up cellular automata (CA) and agent-based models (ABM) are the most widely used models in urban expansion simulation. CA is a dynamical model with discrete time, space, and cell status and consists of four elements: space, cell status, neighbors, and transition rules [8]. CA assumes that historical and neighboring land use states will affect future land mode, so the cellular state transition rules reflect this, and the complex urban spatial structure is simulated through changes to the cell state [9–11]. Therefore, defining the transition rules is the key to CA simulation. The current methods for obtaining transition rules include logistic regression [12], artificial neural network (ANN) [13], random forest [14], and multi-criteria evaluation [15]. However, they still lack standard methods for defining transition rules [16]. Moreover, in a CA model, the spatial location of the cell cannot be moved. It emphasizes the interaction between spatial elements and the surrounding environment and does not explain the process and causes of urban growth [17]. In addition, the urban expansion process involves multiple stakeholders. The interactions of these stakeholders and their decision-making behavior can directly affect urban land conversion, which is difficult for CA to integrate into the model.

Unlike CA, ABM can simulate complex spatial decision-making behavior and stakeholder interactions in land use change [18–22]. In ABM, agents are individuals that exist in and interact with a specific dynamic environment [23,24]. Agents have problem-solving abilities and behavioral goals. They make decisions through shared perceptions and interactions with other agents and represent geographic phenomena in complex spatiotemporal dynamics. Similar to the individual variability of real stakeholders, agents also have different attributes, behaviors, and preferences, which directly affect their decision-making behavior. This results in differences in the location of land use activities [25]. Therefore, defining the agent and obtaining the model's decision parameters is the key to ABM modeling [26,27]. For example, Li et al. [28] selected three main agents (resident, developer, and government) that influence urban development and classified a resident's intelligence based on their income and the presence of children to create an intelligence-based model to simulate the urban expansion of Guangzhou, China. Arsanjani et al. [29] also defined three agent groups (developer, government, and resident) and simulated the spatiotemporal pattern of urban growth in Tehran, Iran, by coupling ABM and multi-criteria analysis models. In addition, Kong et al. [30], Liu et al. [31], and Tian et al. [32] coupled CA and ABM to simulate urban expansion in metropolitan areas such as Beijing and Tianjin, China, from the perspective of residents' location decision-making process.

Overall, the ABM applications for urban locations are often based on residential location selection. Three types of agents (resident, developer, and government) are selected for modeling. Moreover, models mainly use macro or empirical data to obtain the social attributes of agents, such as income (high income, middle income, and low income) and the presence of children based on statistical yearbooks and census information. In contrast, the location attributes of agents are obtained by remote sensing (RS) and geographic information system (GIS) technology. However, the current study has the following concerns.

(1) Due to the agent type, models primarily focus on residential location selection by urban residents and ignore the influence of other decision-makers.

Most ABM-based urban expansion models focus on the behavior of urban agents and tend to ignore the relationship between farmer agent behavior and urban expansion. In China's unique land system, urban land is primarily owned by the state, non-urban

land is collectively owned [33], and urban construction must take place on state-owned land [34,35]. Therefore, the urban expansion precondition is converting collectively owned land into state-owned land through land acquisition. Land acquisition is mainly carried out between the government and farmers, and conflicts between them occur from time to time due to the unequal distribution of benefits. This directly affects the land acquisition process and outcomes, thereby hindering the speed and direction of urban expansion.

Tan et al. [6] and Liu et al. [36] established a land acquisition process model based on agent decisions and adopted Game Theory to resolve conflicts. However, Tan et al. [6] only considered three types of agents: the government, land owners, and land developers, ignoring the critical influence of resident agent locations. Liu et al. [36] did not consider developer agents and used a linear empirical function to define agent behavior rules. At the same time, the game model they built is a unilateral offer model, while the natural land acquisition process is often a bilateral offer. Tang et al. [37] established a bargaining model with fair preferences for both parties (government and farmers) that focuses on securing farmers' benefits and resolving conflicts. This model is a dynamic game of bilateral offer, which can reflect the interaction between the government and farmers in land acquisition and is more suitable for China's land acquisition process. It provides a new way to analyze and quantify agents' decision-making behavior during land acquisition.

(2) Based on the definition of an agent's decision-making behavior, it is difficult for traditional methods to portray its nonlinear relationship with the decision-making environment.

An agent's decision-making behavior occurs in a dynamic geographical environment and there is a complex nonlinear relationship between behavior rules and the decision-making environment. However, most studies use linear functions to define agent behavior rules. They ignore the nonlinear behavior rules of agents who cannot adapt to a dynamic decision-making environment [38]. In addition, the definition of agent behavior rules involves multiple influencing factors, such as MCE [29] and AHP [30,31], and other methods have been widely used to determine model parameters. However, such models based on expert experience are intensely subjective [39], which may affect the accuracy of the simulation.

ANNs are self-adaptive, self-organizing, and have strong learning abilities, so they can fully approximate arbitrary and complex nonlinear relationships [13,40–42]. They are particularly suitable for dealing with nonlinear or complex systems that cannot be described mathematically. For example, Li et al. [13] automatically obtained CA model parameters by training an ANN that was applied to simulate various land use changes. A model using ANN can effectively reflect the complex relationship between spatial variables. Shafizadeh-Moghadam et al. [43], Shafizadeh-Moghadam et al. [44], Tayyebi et al. [45] compared ANN with LR, MCE, and other urban expansion models, and concluded that ANN has more advantages for simulating urban expansion. Therefore, many scholars combine ANN with ABM: Zhao et al. [38] combined ABM, ANN, and an artificial immune system to spatially optimize land use allocation, in which ANN adaptively learned the preferences of land users to express the dynamic and nonlinear characteristics of agent behavior rules. Xu et al. [39] coupled ABM and ANN to reveal the learning process and heterogeneity of multi-sub-residential agents, and the ANN-ABM accurately simulated urban sprawl in Auckland, New Zealand. The above research proves that ANN can more effectively simulate the decision-making behavior of agents in complex systems.

However, few studies have coupled multiple decision processes to simulate urban land expansion, and decision definition needs more intelligent methods. In summary, this study aims to establish an urban expansion simulation model coupled with two sub-models: a residential location selection model and a land acquisition bargaining model. Those two sub-models contain four types of agents: resident agents (RA), real estate developer agents (DA), government agents (GA), and farmer agents (FA). The two sub-models have the following main characteristics: (1) residential location selection (RA, DA, and GA) and land acquisition bargaining (GA and FA) are used to simulate urban expansion; (2) ANN is used to define the behavior rules of RA to obtain the nonlinear relationship and reduce

subjective effects when defining parameters; and (3) a bilateral bargaining model with fair preference is used to simulate the land acquisition process between GA and FA.

## 2. Methodology

The coupled model includes a residential location selection sub-model and a land acquisition bargaining sub-model; the former consists of the decision-making process of RA, DA, and GA. First, RA selects possible residential locations based on personal preferences, and then the DA decides whether to develop the land based on income. It will apply to the GA for land use if it is developed. The GA determines whether to approve land development based on planning and other urban expansion conditions. If approved, it will enter the land acquisition bargaining sub-model; otherwise, the land application shall be rejected. In the second model, GA and FA carry out dynamic bilateral bargaining on the land. When the income obtained by the two parties reaches their respective expectations, both parties agree to expropriate the land. If the land acquisition is successful, it will be converted into urban land; otherwise, it will not be converted. Therefore, this study expresses the land urbanization results as follows:

$$U \sim \left( f_{sel}, f_{acq} \right) \tag{1}$$

where $U$ is the result of urban expansion, $f_{sel}$ is the residential location selection sub-model result, and $f_{acq}$ is the land acquisition bargaining sub-model result.

### 2.1. Residential Location Selection Sub-Model

2.1.1. The Behavior of the Resident Agent

The residential location selection decision is an important driving factor of urban land expansion that reflects the spatial component of urban residential housing behavior. In general, RAs choose residential settlements in specific zones based on their needs in terms of housing price, transportation convenience, living environment, and living convenience. Different RAs express spatial location preferences based on their attributes, with different spatial decision-making behaviors. This study used ANN to obtain the relationship between new residential land and driving factors. The RA residential preference was expressed by the selection probability, which is calculated as follows:

$$Sel_{ij} = f_{\text{ANN}}(x_1, x_2, \cdots, x_n) \tag{2}$$

where $Sel_{ij}$ denotes the probability of RA site selection candidate location $L_{ij}$, $f_{\text{ANN}}$ is the ANN function, and $x_n$ denotes the factors affecting RA site selection.

Multilayer perceptron (MLP) is the most commonly used type of neural network, and it contains input, hidden, and output layers (Figure 1). When ANN is applied in urban simulation research, the input layer is the spatial variable affecting urban expansion. The output layer is the result of urban land change. This model randomly selects data to train the ANN and then applies the trained ANN (optimal weights) to the whole dataset to obtain a RA site selection probability map.

The biggest drawback of an ANN is its "black box" aspect, meaning the user may not clearly understand the modeling mechanisms, especially the contribution of the input variables to the predicted output values. However, the relationship between urban expansion and its driving factors is one of the most concerning topics in urban modeling. The Garson–Goh algorithm [46–48] is widely used in ANN-related research and partitions the neural network link weights to determine the relative importance of each input variable in the network. It was used in this study to quantify the relative contribution of the independent variable (driver) to the dependent variable (RA residential preference). The algorithm includes the following main steps:

First, the contribution of each input layer node to the output layer node is calculated through the hidden layer nodes:

$$C_{ij} = W_{ij} \times W_{jk} \tag{3}$$

where $W_{ij}$ denotes the link weight of input layer node $i$ to hidden layer node $j$; $W_{jk}$ denotes the link weight of hidden layer node $j$ to output layer node $k$.

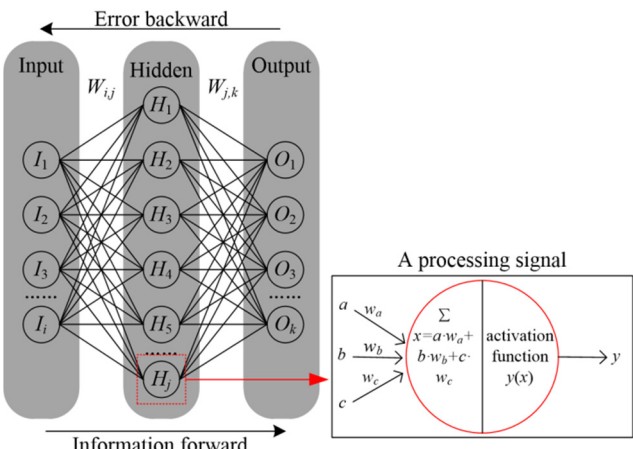

**Figure 1.** Artificial neural network architecture.

Then, the relative contribution of each input layer node to the input signal of the hidden layer node is calculated:

$$R_{xy} = |C_{xy}| / (|C_{1y}| + |C_{2y}| + \cdots + |C_{ij}|) \tag{4}$$

The sum of the input layer node contributions is calculated:

$$S_x = R_{x1} + R_{x2} + \cdots R_{xj} \tag{5}$$

Finally, the relative importance of each input variable is calculated:

$$RI_x = S_x / (S_1 + S_2 + \cdots S_i) \times 100\% \tag{6}$$

2.1.2. The Behavior of the Real Estate Developer Agent

The DA plays an essential role in urban land expansion by applying to the GA for land for real estate development activities. The RA's housing preference and the GA's land management policy jointly affect the DA's decision-making behavior. First, the DA must evaluate the location of the land to be developed and the surrounding environment (socioeconomic and natural environment) and make investments based on the RA's housing preferences to ensure that houses are easy to sell. Second, the DA's development activities need to consider development costs to make a profit. The profit calculation formula for a DA is as follows:

$$D_{profit} = H_{price} - L_{price} - D_{cost} \tag{7}$$

where $D_{profit}$ denotes the profit of DA investment, $H_{price}$ is the house price, $L_{price}$ is the land price, and $D_{cos}$ is the development cost.

The DA is simplified due to the difficulty in obtaining development costs and investment profits. Due to cost and risk considerations, DAs tend to choose areas for investment with complete infrastructure and RA concentrations, and the infrastructure condition is also the most crucial concern for most home buyers. Therefore, the study used neighborhood impacts as a metric for DA's willingness to develop:

$$Dev_{ij} = \frac{\sum\limits_{3 \times 3} N(\text{urban}_{ij})}{3 \times 3 - 1} \tag{8}$$

where $Dev_{ij}$ is the probability that a DA chooses to develop at the candidate position $L_{ij}$ and $\sum\limits_{3\times3} N(\text{urban}_{ij})$ is the number of urban pixels in a $3 \times 3$ neighborhood. If $\sum\limits_{3\times3} N(\text{urban}_{ij}) = 0$, $Dev_{ij} = 0.05$.

### 2.1.3. The Behavior of the Government Agent

The primary responsibility of the GA is macro urban planning and regulation, guiding a city's evolution process and determining the development mode. A DA evaluates the benefits and risks of land development and decides whether or not to develop. It will apply to the GA for land use if it chooses to develop. The GA decides whether to approve the application according to whether the land aligns with urban planning. A GA's decision-making behavior is as follows:

$$Reg_{ij} = \left\{ \begin{array}{l} 1, \text{ Outside the prohibited development zone} \\ 0, \text{ Within the prohibited development zone} \end{array} \right. \tag{9}$$

where $Reg_{ij}$ is the decision probability for a GA development application at candidate location $L_{ij}$. A binarized layer represents the GA decision behavior. When the value is 1, $L_{ij}$ is a permitted development area in urban planning, and the GA agrees to DA's application for land development. When the value is 0, $L_{ij}$ is a prohibited development area, and the GA does not agree to the application.

The total probability of the resident location selection sub-model is jointly determined by RAs, DAs, and GAs, which can be expressed as follows:

$$Set_{ij} = Sel_{ij} \times Dev_{ij} \times Reg_{ij} \times \lambda \tag{10}$$

where $Set_{ij}$ is the total probability of resident location selection at candidate location $L_{ij}$, $\lambda$ is the adjustment parameter of the model, $\lambda \in [0,1]$.

### 2.2. Land Acquisition Bargaining Sub-Model

In a two-party bargaining model with fair preferences, GA and FA take turns making offers for expropriated land, and the GA acts first. The GA decides whether to acquire land based on urban construction needs and the proceeds of land acquisition. If the land is acquired, the GA makes an initial offer based on the land acquisition compensation standard. The FA has three choices: accept (GA's offer is higher than personal expectations), reject (the offer is lower than the compensation standard), and counter-offer (the offer is higher than the compensation standard but lower than personal expectations). The first two decision-making behaviors constitute the first round of bargaining, and a counter-offer enters the second round. The FA will make a counter-offer based on the individual's initial expectations. Similarly, the GA has three options: accept (profitable and above expectations), reject (unprofitable), and counter-offer (profitable but below expectations), with the first two decision-making behaviors constituting the second round and a counter-offer entering the third round. The GA and FA take turns bargaining indefinitely, ending each round of bargaining when one side accepts or rejects the other's offer, with acceptance representing successful land acquisition and rejection failing.

Fairness psychology affects the decision-making behavior of bargaining participants in land acquisition. Because they are driven by fairness psychology, the GA and FA will sacrifice their individual interests to some extent to reach an agreement. Moreover, the land acquisition process is time-consuming, and each round of bargaining incurs costs (denoted as the discount coefficient $\delta$, $0 < \delta < 1$), which motivates the bargaining parties to reach an agreement as soon as possible to reduce losses. During the bargaining process, the GA and FA have their strategies and are influenced by each other. Based on their different strategies, the benefits of different rounds can be calculated, and the land acquisition result can be evaluated. For details on GA and FA strategies during the bargaining process and the consideration of fairness psychology (see [37]).

In the model, for odd rounds, the GA offers, and FA makes the choices; for even rounds, the FA offers, and GA makes the choices. Therefore, there are two cases for GA and FA benefits:

$$
R_{ij}^n = \begin{cases} \delta_{n-1}^f \times C_{ij}^n & \text{FA accepted the GA's offer} \\ \delta_{n-1}^f \times F_{ij}^n & \text{GA accepted the FA's offer} \end{cases} \tag{11}
$$

$$
G_{ij}^n = \begin{cases} \delta_{n-1}^g \times \left[ M_{ij} \times (1-e) - C_{ij}^n \right] & \text{FA accepted the GA's offer} \\ \delta_{n-1}^g \times \left[ M_{ij} \times (1-e) - F_{ij}^n \right] & \text{GA accepted the FA's offer} \end{cases} \tag{12}
$$

where $R_{ij}^n$ and $G_{ij}^n$ are the FA and GA returns at candidate position $L_{ij}$ in the $n$th round, respectively; $\delta_{n-1}^f$ and $\delta_{n-1}^g$ are the FA and GA discount factors, respectively; $C_{ij}^n$ is the GA offer price in round $n$ ($n$ is an odd number); $F_{ij}^n$ is the FA offer price in round $n$ ($n$ is an even number); $M_{ij}$ is the modified GA land transfer price; and $e$ is the coefficient of land development cost.

Because bargaining participants are rational and there is upper-level government supervision, the bargaining rounds cannot be infinite. The final result may be that the GA gives the final offer in a particular round, which the FA decides. If the GA's offer is higher than the FA's expectation in that round, the FA agrees to the land acquisition; otherwise, the land acquisition fails. The GA and FA's expected revenue is dynamically changing. At the same time, the ratio of realistic land acquisition GA and FA gains are about 60~70% and 20~40% [49], based on the assumed minimum and maximum expectations of the bargaining parties. The minimum expectation is accepted by the GA and FA only when the final round of bargaining is conducted; that is, the final GA offer is based on the minimum expectation of the individual, and the expectation of FA in the final round is the minimum expectation. This study assumed that the two parties conduct five rounds of bargaining, and the final land acquisition result was determined as follows:

$$
Acq_{ij} = \begin{cases} 1, & G_{ij}^5 \geq E_{ij}^g \text{ and } R_{ij}^5 \geq E_{ij}^f \\ 0, & \text{other} \end{cases} \tag{13}
$$

where $Acq_{ij}$ is the land acquisition result of candidate position $L_{ij}$, 1 denotes the land acquisition success, and 0 denotes the land acquisition failure. $E_{ij}^g$ and $E_{ij}^f$ are the expected GA and FA revenues, respectively.

### 2.3. Urban Expansion Simulation Coupled with Residential Location Selection and Land Acquisition Bargaining

Only residential land expansion was considered in this study, so it was assumed that all urban land was for residential use. It was assumed that each urban pixel could accommodate one RA. The number of RAs was determined based on the total urban land use growth. When each RA selected a residential location, the model first calculated the probability $Sel_{ij}$ of a housing location for each pixel (except those selected by a specific RA). It then sorted the pixels in descending order of probability. The pixel with the highest probability entered the RA, DA, and GA joint decision-making process. The highest pixel of joint decision development probability $Set_{ij}$ (selected by roulette when there are multiple pixels with the same probability) was entered into the land acquisition bargaining sub-model as the candidate RA location. If the land acquisition was successful, the pixel was transformed into urban land and occupied by the RA; otherwise, the RA continued to select the next pixel with the highest probability and entered the bargaining process again. The model iterated in this way until the RA found a suitable location to settle. Each iteration

of the model represented an RA selection process (Figure 2). Based on this, the urban expansion probability in this model was expressed as follows:

$$P_{ij} = Set_{ij} \times Acq_{ij} \tag{14}$$

where $P_{ij}$ is the urban expansion probability of candidate location $L_{ij}$.

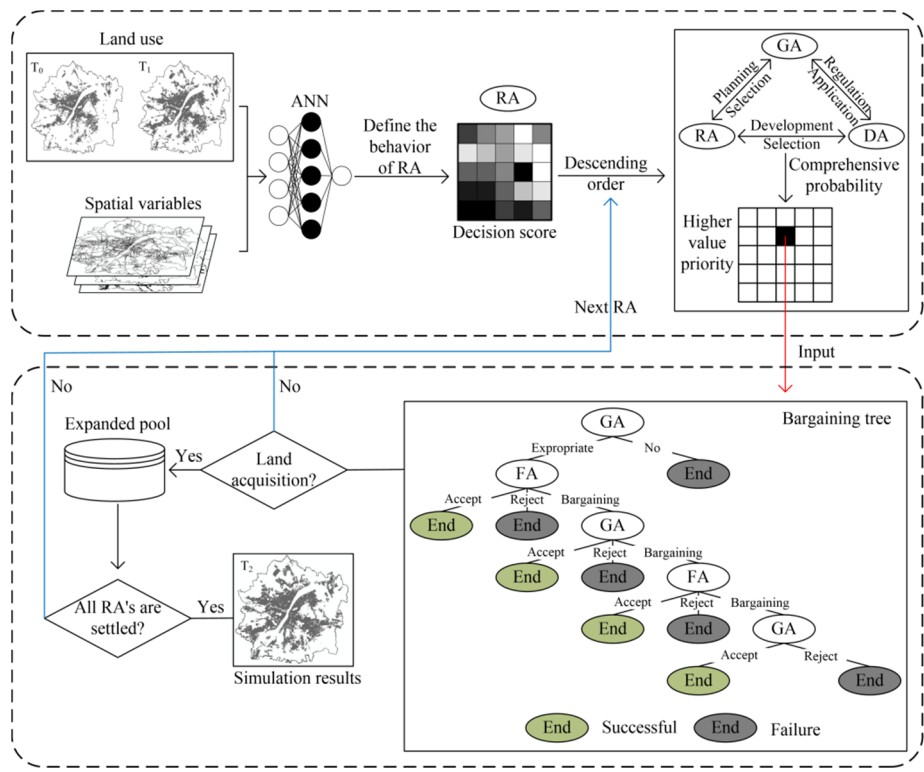

**Figure 2.** Model flow chart.

## 3. Research Area and Data

### 3.1. Research Area

Wuhan is the capital of Hubei Province in central China, the core city of the Yangtze River Economic Belt, and the central region's most prominent political and economic center. With the implementation of the Rise of Central China strategy and the construction of Wuhan's Urban Circle, Wuhan's economy has developed rapidly. In 2021, the GDP of Wuhan reached CNY 1771.68 billion, an increase of 12.2% over the previous year. At the same time, the city's permanent urban population at the end of the year was 11.54 million, and the urbanization rate was 84.56%. As a result of economic and population growth, Wuhan has experienced a rapid urban expansion in the past decades, with an increase in built-up area from $4.19 \times 10^4$ ha in 1988 to $43.39 \times 10^4$ ha in 2011 [14].

The administrative division of Wuhan includes seven central urban districts: Jiang'an, Jianghan, Qiaokou, Hanyang, Qingshan, Wuchang, and Hongshan, and six distant urban areas: Huangpi, Xinzhou, Jiangxia, Hannan, Caidian, and Dongxihu. The total land area is about 8569.15 km$^2$. In this study, the urban development zone (WHUDZ) defined by the urban planning of Wuhan was selected as the experimental area (Figure 3), which is the main gathering area for urban functions and the critical expansion area for urban space in Wuhan. It has a total land area of about 3261 km$^2$.

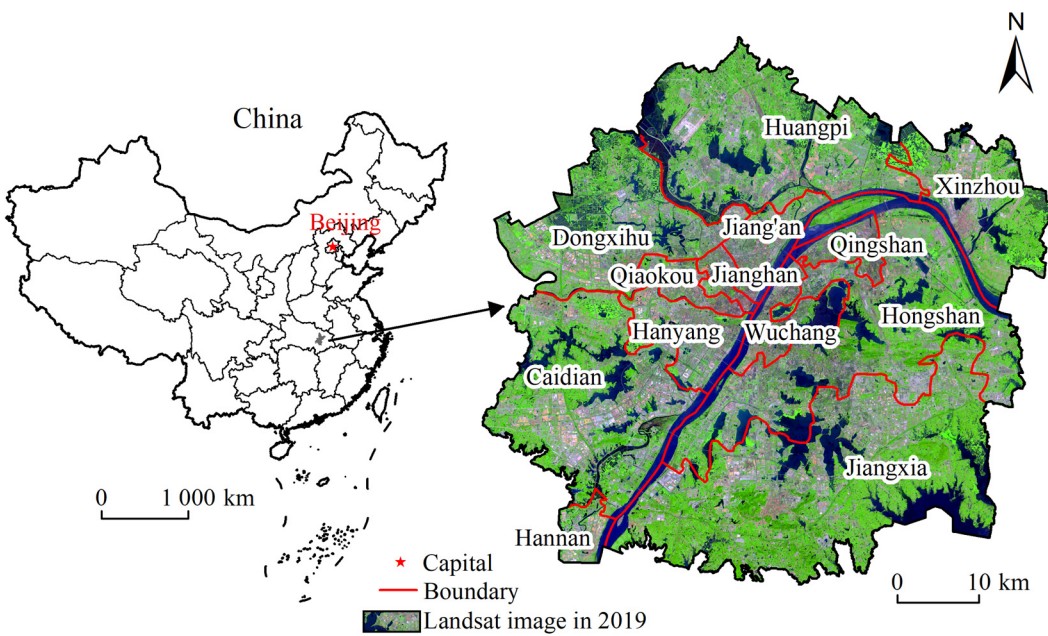

**Figure 3.** Location and remote sensing image of the study area.

*3.2. Data Preparation and Processing*

The spatial and socioeconomic data used in the study are listed in Table 1. Landsat TM/OLI images in 2009, 2014, and 2019 were used to obtain land-use change information. After preprocessing, the remote sensing images were classified into five categories using maximum likelihood classification: cultivated land, forest land, water bodies, urban construction land, and other lands. Three hundred sample points were randomly selected to evaluate the classification results. The results showed that the classification accuracy of each period was greater than 85%, meeting research needs. Two land use types, forest land, and water bodies were reclassified as restricted development zones, and cultivated land and other land use were reclassified as non-urban land. In addition, the data were resampled to 100 m to reduce the simulation's computational complexity and running time.

**Table 1.** Data used in this study.

| Type | Name | Year | Data Source |
|------|------|------|-------------|
| Spatial data | Landsat TM and OLI | 2009, 2014, and 2019 | USGS |
| | DEM | | USGS |
| | Road network | 2015 | China Basic Geographic Database (1:250,000), Open Street Map |
| | Gaode POIs | 2019 | Gaode map (https://ditu.amap.com/), accessed on 1 August 2020 |
| Socioeconomic data | GDP and population density | 2015 | RESDC (http://www.resdc.cn), accessed on 1 August 2020 |
| | House price | 2019 | Lianjia (https://wh.lianjia.com), accessed on 1 August 2020 |
| | Land acquisition compensation standard | 2019 | Department of Natural Resources of Hubei Province |
| | Benchmark land price for residential | 2019 | Wuhan natural resources and planning bureau |

With reference to related research and data availability [28,31,42,50], 14 factors were selected, including DEM, slope, distance to major roads, distance to expressways, distance to bus stations, distance to subway stations, house price, distance to parks, distance to water sources, distance to schools, distance to hospitals, distance to supermarkets, GDP

and population density, etc., to analyze the influencing factors of RA residential location selection considering the terrain, transportation, housing prices, and convenience (Figure 4a–n). The data were normalized to [0,1]. The housing price data were from the housing price monitoring information of Wuhan City in 2019, released by Lianjia. A total of 2221 monitoring points in the study area were obtained, and a regional housing price layer was generated through the Thiessen polygon. A distance layer was generated using Euclidean distance analysis, and the benchmark land price and land acquisition compensation standard layers required by the bargaining model were obtained through vectorization (Figure 4o–p). Data processing and analysis were completed using ENVI 5.3, ArcGIS 10.4, and Python 3.7. The ANN was trained in MATLAB using Neural Network Toolbox. All data had the same spatial reference (UTM WGS 84) and resolution (100 m).

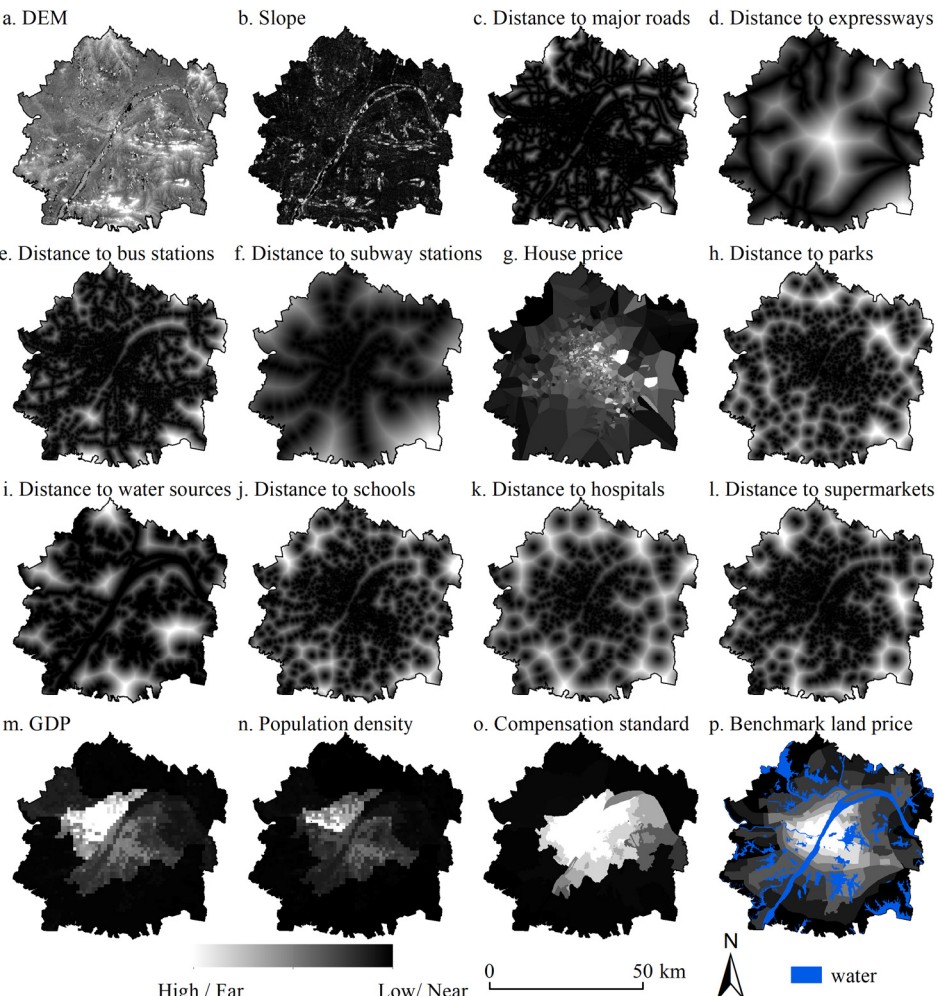

**Figure 4.** Spatial variables influencing urban expansion in the WHUDZ.

## 4. Model Implementation and Results

First, the ANN was trained using land use data from 2009 and 2014 and 14 spatial variables to generate an RA site selection probability map ($Sel_{ij}$). Then, based on the current land use in 2014, two sub-models of residential location selection and land acquisition bargaining were run to simulate the urban expansion of WHUDZ in 2019. This was compared with actual urban land in 2019 to evaluate the model's accuracy. After the simulation accuracy met the requirements, data from 2014 to 2019 were used to retrain the ANN and generate a new RA location probability map. Finally, the urban expansion of WHUDZ in 2029 was predicted based on the 2019 land use status.

### 4.1. RA Location Selection Probability Using ANN

The ANN structure constructed in this study was 14-12-1, meaning there were 14 input layers, 12 hidden layers, and 1 output layer. The input layer nodes consisted of 14 spatial variables that affected the location of RA residences. When building the network, care should be taken when selecting the number of hidden layer nodes. If the number is too small, the network prediction error will be significant, but if too many nodes lead to increased learning time, the phenomenon of "over-fitting" may occur. Wang [51] argued that for a three-layer neural network, the number of nodes in the hidden layer should be at least 2/3 of the number of nodes in the input layer. After several trials, the number of nodes in the hidden layer was set to 12. This study only considered urban land conversion, and the output layer was 1. The data labels were 0 and 1, indicating unconverted and converted to urban land, respectively.

In the urban land conversion map from 2009 to 2014 and 14 spatial variable maps, 30,000 points were selected by random sampling, 15,000 points each for unconverted and converted to urban land. Of those, 70% were used for ANN training, 15% were used as the validation dataset, and 15% were used as the test dataset. The initial weights were generated randomly during ANN training, and the weight update algorithm used "trainlm" with a learning rate of 0.01. The number of epochs was 500, and the mean-square error (MSE) target for stopping training was 0.01. After the ANN training was complete, the optimal weights were used to generate the RA location selection probability map (Figure 5).

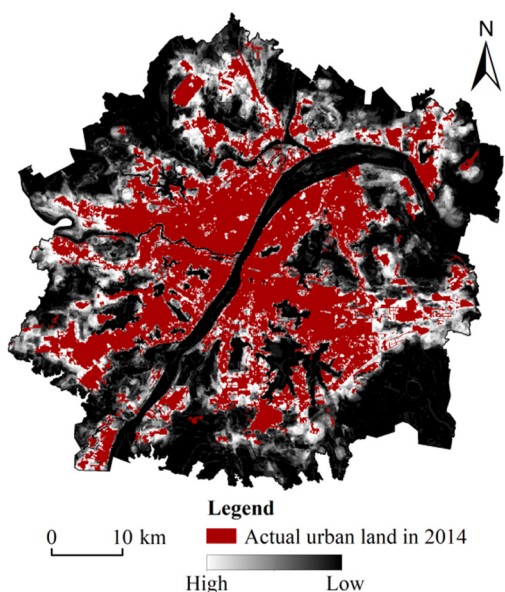

**Figure 5.** The RA location selection probability map.

### 4.2. Urban Expansion Simulation Dynamic Process and Results in 2019

Urban land in WHUDZ increased from 648.37 km$^2$ in 2009 to 889.56 km$^2$ in 2014 and further increased to 1086.15 km$^2$ in 2019, a rapid urban expansion rate that was much higher than the predicted urban planning of Wuhan. Based on land use data for 2009 and 2014, the Markov chain (MC) was used to predict the urban land area of WHUDZ in 2019 to be 1099.91 km$^2$, which was 13.76 km$^2$ more than the actual urban land area with a relative error of 1.27%. The total number of RAs was finally determined to be 21,025.

The coupled model was used to simulate WHUDZ urban expansion from 2014 to 2019, and the dynamic process is shown in Figure 6. N represents the number of resettled residents (the number of pixels increased by urban land), N = 0 indicates that the model is in the initial state representing 2014 urban land, and N = 21,025 indicates that the model has iterated 21,025 times and reached the termination state that represents the simulated 2019 urban land use. During the simulation process, pixels with high development proba-

bility and a large number of urban pixels in the neighborhood were first transformed into urban land. In the early stage, it was mainly characterized by urban filling expansion. In the later stage, it was mainly an extensional urban expansion mode with the continuous expansion of the city scope.

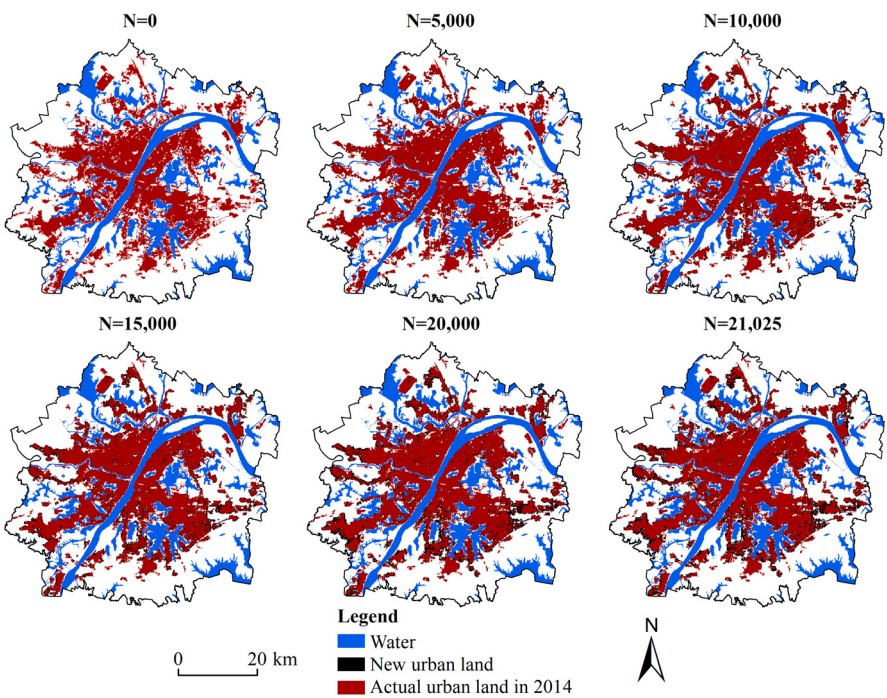

**Figure 6.** Dynamic simulation of urban expansion in the WHUDZ.

*4.3. Accuracy Evaluation and Comparison of Urban Expansion Simulation Results in 2019*

The simulation results were compared with actual land use in 2019 pixel by pixel after eliminating the initial urban land (2014). The accuracy of the simulation was assessed using the Kappa coefficient. Moreover, logistic regression CA (LRCA) was selected in GeoSOS software [52] to simulate the WHUDZ urban land in 2019 and compare the accuracy of the two models using the same dataset. Table 2 shows that with an overall accuracy and Kappa coefficient of 92.78% and 55.24%, respectively, the coupled model developed in this study had excellent simulation accuracy and could successfully simulate WHUDZ urban expansion. The accuracy of the model constructed for this paper was higher than that of LRCA, particularly for urban land simulation. The overall accuracy and Kappa coefficients were 0.76% and 4.68% higher, respectively.

**Table 2.** Error matrix for simulated and actual urban land in the WHUDZ in 2019.

|  | Land-Use Type | Actual | | Simulation Accuracies/% | | | |
|---|---|---|---|---|---|---|---|
|  |  | Urban | Non-Urban | Producer's Accuracy | User's Accuracy | Overall Accuracy | Kappa Coefficient |
| Coupled model | Urban | 12,042 | 8983 | 61.25 | 57.27 | 92.78 | 55.24 |
|  | Non-urban | 7617 | 201,116 | 95.72 | 96.35 |  |  |
| LRCA | Urban | 11,175 | 9850 | 56.84 | 53.15 | 92.02 | 50.56 |
|  | Non-urban | 8484 | 200,249 | 95.31 | 95.94 |  |  |

In addition, five landscape metrics were selected to assess the similarity between the simulation results and the actual pattern (2019), including the number of urban patches (NP), largest-patch index (LPI), mean Euclidean nearest-neighbor distance (ENN_MN), mean perimeter-area ratio (PARA_MN), and the proportion of like adjacency (PLADJ).

Table 3 shows that correlations were consistent between the two model simulations and the actual landscape pattern metrics.

**Table 3.** Similarity between simulated results and real landscape patterns.

| Landscape Metrics | NP | LPI | ENN_MN | PARA_MN | PLADJ |
|---|---|---|---|---|---|
| Observed 2019 | 202 | 39.91 | 250.78 | 363.44 | 90.09 |
| LRCA | 128 | 40.91 | 215.13 | 519.07 | 93.17 |
| Coupled model | 150 | 40.55 | 235.35 | 478.41 | 92.73 |

The simulation results showed that LPI, PARA MN, and PLADJ were more significant than the actual pattern. At the same time, NP and ENN MN values were lower than the actual pattern, indicating that the simulated urban landscape pattern was more fragmented and the urban form was more compact. The primary similarity errors all came from PARA MN. All five landscape indices in the linked model were more comparable and closer to the real pattern than those of the LRCA model, demonstrating that the coupled model more accurately simulated urban development pattern evolution in the WHUDZ.

Figure 7 shows WHUDZ urban expansion as simulated by the coupled model in 2019 compared with the actual urban land. Missing values indicate actual expansion, while the simulation was not expanded. These were mainly located in the periphery of the city, most of which were adjacent to the original urban land, and some were far from the original urban land. As shown in Figure 7a, the missing pixels were clustered in this area. The main reason was that the urban expansion was not apparent in the past, the road density was low, and the RA location probability obtained by ANN training was low, indicating that the coupled model poorly simulated the expansion of the peripheral area. Hits represented both actual and simulated expansion, mainly filled expansion pixels within the city and the conversion of land with high traffic accessibility. For example, in Figure 7b, located in the Wuhan East Lake High-tech Development Zone, the urban land increased significantly, the complete infrastructure, the urban land conversion probability was high, and there were more hits. This zone may continue to be a hot spot for urban expansion. False hits were mainly distributed in areas with a high probability of urban expansion, such as Figure 7c, which experienced rapid urban land expansion from 2009 to 2014 and a high probability of urbanization of surrounding sites. These were more likely to have simulated expansion without actual expansion. Overall, the coupled model accurately simulated 57.27% of the newly added urban pixels in WHUDZ from 2009 to 2014, with higher simulation accuracy and better performance than the traditional CA model.

*4.4. Prediction of Future Urban Expansion in the WHUDZ*

Based on land use data from 2009 to 2019, MC predicts that by 2029, WHUDZ urban land will reach 1415.82 km$^2$, an increase of 329.67 km$^2$ compared to 2019 and twice the area of urban land in 2009. By taking 2019 as the starting point for land use, the simulated WHUDZ urban land in 2029 is shown in Figure 8. The spatial characteristics of WHUDZ's prospective urban development demonstrate that it generally spreads outward around the original urban land and expands in all directions, with the southeast and northwest being the most active growth regions. The landscape metric shows that the predicted urban spatial pattern is more compact, and the spatial structure is optimized. The city's link with nearby metropolitan areas will deepen due to the city's ongoing urban land expansion, and Jiangxia, Caidian, and Huangpi districts have the most growth.

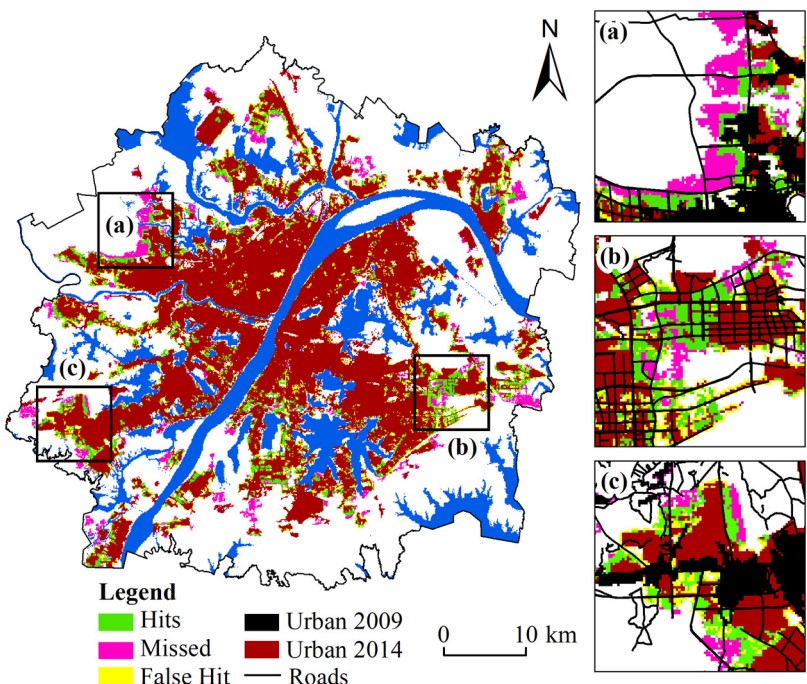

**Figure 7.** Comparison of simulation results and a reference map for urban expansion in the WHUDZ in 2019.

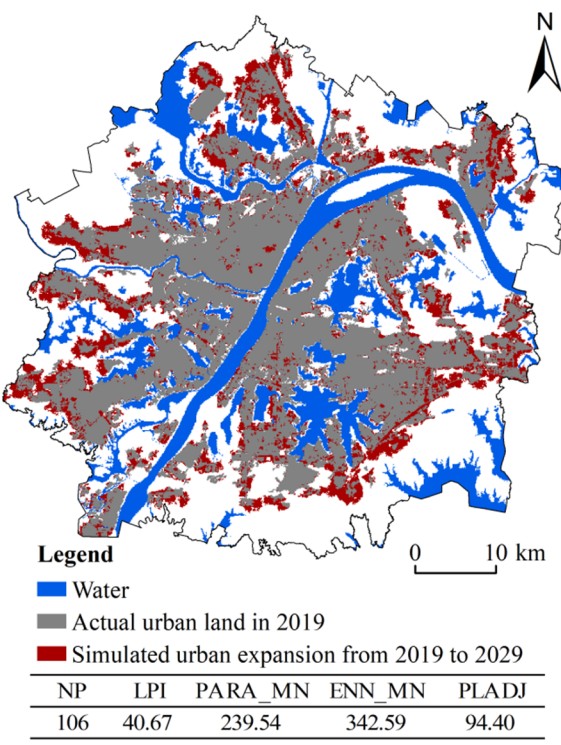

| NP | LPI | PARA_MN | ENN_MN | PLADJ |
|----|-----|---------|--------|-------|
| 106 | 40.67 | 239.54 | 342.59 | 94.40 |

**Figure 8.** Predicted urban land use in the WHUDZ in 2029.

## 5. Discussion

### 5.1. The Significance of a Coupled Residential Location Selection and Land Acquisition Bargaining Model

The urban system is very large and complex, with spatial–temporal dynamic changes. From the microscopic point of view, urban land use changes result from human activities within an urban system in the geographic environment. China has a unique land system

in which urban construction is often carried out through large-scale expropriation of collective land used by farmers, mainly arable land. The government expropriates land for infrastructure construction and then sells the land use rights according to specific uses determined by urban planning. Residential land is one of the most widely distributed types of urban land. Land developers obtain development permits from government departments and then develop real estate for urban land conversion, which is influenced by residents' demand for housing. The decision-making behavior of the government, farmers, developers, and residents runs through the process of converting non-urban land to urban land in China.

This study simulated urban expansion by coupling residential location selection (RA, DA, and GA) and land acquisition bargaining (GA and FA) to reveal the impact of each agent's decision-making behavior on urban expansion. In the real world, the joint effect of these agents on the conversion of non-urban land to urban land is much more pronounced than the impacts of natural and spatial elements. The sub-model initially simulated residential living locations, RAs selected suitable residences in accordance with their own needs, DAs developed land and built real estate to seek maximum profits, and GAs complied with urban planning to sell land use rights. Those three types of agents were independent and interacting.

A bargaining model simulated the process of land acquisition between GAs and RAs. It was assumed that the land chosen by a RA could not be converted into urban land if land acquisition failed in the sub-model. The main distribution of such pixels is shown in Figure 9. In the 2019 WHUDZ urban land simulation process, 1324 pixels could not be converted into urban pixels due to unsuccessful land acquisition, accounting for about 6.3% of the total simulated pixels. There were two main situations in which land acquisition was unsuccessful. Type 1 indicated that the government and farmers were unwilling to carry out land acquisition activities, totaling 878 pixels. In contrast, type 2 indicated that the government was willing to acquire land, but farmers were unwilling to sell, with a total of 446 pixels.

As shown in Figure 9a,c, type 1 was mostly located in the periphery of the city, especially in areas far from the center where government compensation standards for land expropriation and the benchmark land price were both low. Because both the GA and FA received modest gains from land expropriation, it was difficult to have successful land acquisition because neither side wanted to do it. As shown in Figure 9b, type 2 was mostly located around the city center with high urbanization probability, high value-added land, and low compensation standard for land expropriation. A GA could make a significant amount of profit from land expropriation, whereas a FA made very little. The FA would become reluctant to participate, and the land expropriation was prone to failure. It is worth pointing out that the income of those two types of agents is quite different because the land expropriation compensation standard and the benchmark land price for type 2 were quite different (Figure 4o,p). In addition, some pixels with high $Set_{ij}$ (such as pixels (247, 558) and (398, 333)) could not be converted into urban land because the land acquisition was unsuccessful.

This study combined residential location selection and the land acquisition process, which mainly affect urban land conversion in China, to describe the urban land development process. It also took the main agents' decision-making behavior into account with respect to the overall macroscopic situation of urban development. When compared with the CA model, the coupled model obtained higher simulation accuracy and a landscape pattern closer to actual land use.

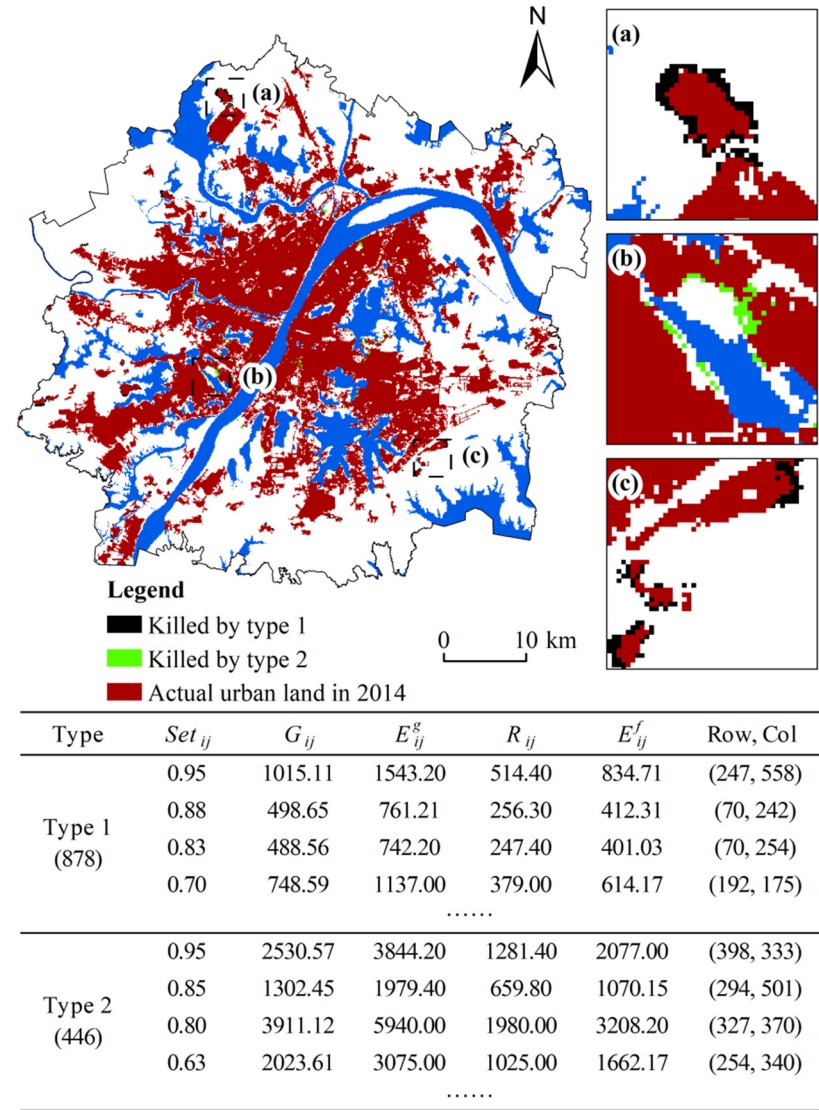

| Type | $Set_{ij}$ | $G_{ij}$ | $E_{ij}^g$ | $R_{ij}$ | $E_{ij}^f$ | Row, Col |
|---|---|---|---|---|---|---|
| Type 1 (878) | 0.95 | 1015.11 | 1543.20 | 514.40 | 834.71 | (247, 558) |
| | 0.88 | 498.65 | 761.21 | 256.30 | 412.31 | (70, 242) |
| | 0.83 | 488.56 | 742.20 | 247.40 | 401.03 | (70, 254) |
| | 0.70 | 748.59 | 1137.00 | 379.00 | 614.17 | (192, 175) |
| | | | …… | | | |
| Type 2 (446) | 0.95 | 2530.57 | 3844.20 | 1281.40 | 2077.00 | (398, 333) |
| | 0.85 | 1302.45 | 1979.40 | 659.80 | 1070.15 | (294, 501) |
| | 0.80 | 3911.12 | 5940.00 | 1980.00 | 3208.20 | (327, 370) |
| | 0.63 | 2023.61 | 3075.00 | 1025.00 | 1662.17 | (254, 340) |
| | | | …… | | | |

**Figure 9.** Land acquisition failed in the bargaining and could not be transformed into urban land (Notes: Killed by type 1 means that neither the government nor farmers are willing to expropriate land. Killed by type 2 means the government wants to expropriate land, and farmers are unwilling to sell).

### 5.2. Factors Influencing RA Residential Location Selection

In this study, 14 variables were selected to analyze the factors influencing RA residential location, ANN was used to obtain the nonlinear relationship between them, and the Garson–Goh algorithm was used to reveal the relative contribution of ANN input variables to output variables (Figure 10). Traffic accessibility (distance to major roads, distance to subway stations, and distance to bus stations) generally contributed significantly to RA residential location selection decisions. Among those factors, distance to major roads had the greatest impact, indicating that land closer to the main road was more likely to be selected by the RA as residential and thus converted to non-urban land, which was consistent with the research conclusion of Tan et al. [35]. Despite being a relatively minor factor in RA location selection, the importance of amenities such as distance to parks, water sources, schools, hospitals, and supermarkets has been growing over time. This is mainly because, as cities have rapidly expanded and infrastructure such as roads has improved, commute times and distances are less, and lifestyles have changed. This trend is likely to continue in the future.

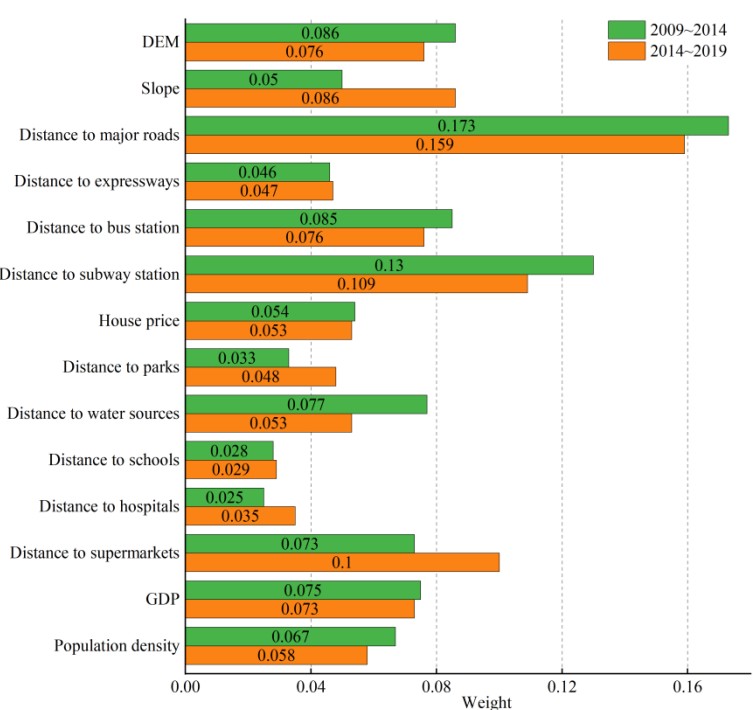

**Figure 10.** Contributions of different influencing factors.

*5.3. Changes to GA and FA Income before and after Bargaining*

The two-party bargaining model with fair preference coordinated conflicts of interests between the GA and FA in the land acquisition process to ensure the smooth implementation of land acquisition activities. In the areas where land acquisition bargaining was successful, FA's gains were improved, and the GA made appropriate profits and reached an agreement that may benefit both parties. Conflicts did not occur or were less likely, so urban expansion could be carried out within this range to reduce land acquisition conflicts. As shown in Table 4, the average revenue of the FA in land expropriation increased from 306.3 CNY m$^{-2}$ to 604.76 CNY m$^{-2}$, with a net increase of 97.44% from 2014 to 2019 when new simulated urban land pixels were added. GA average revenue decreased from 2099.87 CNY m$^{-2}$ to 1624.48 CNY m$^{-2}$, a net decrease of 22.64%. Through bargaining, FA income increased significantly, and the income gap between GA and FA decreased, but the GA still obtained most of their income from land expropriation.

**Table 4.** Changes in income of government and farmers before and after bargaining (Unit: CNY m$^{-2}$).

|  | **Before** | **After** | **Growth/%** |
|---|---|---|---|
| FA | 306.30 | 604.76 | 97.44 |
| GA | 2099.87 | 1624.48 | −22.64 |

## 6. Conclusions

This study simulates urban expansion using two coupled sub-models: residential location selection and land acquisition bargaining, which consisted of four agents: RA, DA, GA, and FA. First, RA, DA, and GA. ANN was used to define the RA behavior rules, which selected settled pixels according to the probability generated by the land acquisition bargaining sub-model composed of GA and FA. Both agents decided whether to expropriate the land according to their respective returns. If the land acquisition was successful, the pixel was converted to urban land; otherwise, it was not. Each iteration represented the RA selection process.

WHUDZ was selected for this empirical study. The urban land expansion was predicted based on WHUDZ data from 2009 to 2014. The results showed that the total accuracy

and Kappa coefficient of land use in 2019 were 92.78% and 55.24%, respectively. Moreover, the LRCA simulation results were 92.02% and 50.56%, respectively. The accuracy of the coupled model was higher than LRCA, which demonstrated the model's effectiveness. The coupled model predicted that the urban land area of WHUDZ will reach 1415.82 $km^2$ in 2029, primarily indicating outward expansion within the southeast and northwest regions. In addition, the Garson–Goh algorithm was used to reveal the relative contributions of various influencing factors in ANN to the RA residential location choices. It was found that the overall contribution of traffic accessibility was more significant, distance to major roads had the greatest impact, and the weight of life conveniences gradually increased with time. Moreover, this trend is likely to continue. In the simulation of WHUDZ's 2019 urban land use, the land acquisition bargaining sub-model killed a total of 1324 pixels, which could not be converted into urban pixels after land acquisition failed.

In this study, two crucial decision-making processes affecting urban land expansion in China are coupled, and the urban evolution of WHUDZ is simulated to reflect the impact of micro-individual decisions on global urban growth. In particular, quantifying the land expropriation process is relevant for solving land expropriation conflicts and formulating more reasonable land expropriation policies.

**Author Contributions:** Conceptualization, writing—original draft preparation, methodology, software, visualization, H.L.; validation, formal analysis, data curation, writing—review and editing, L.Z.; supervision, project administration, funding acquisition, D.T. All authors have read and agreed to the published version of the manuscript.

**Funding:** This research was funded by the National Natural Science Foundation of China, grant number 71663017.

**Institutional Review Board Statement:** Not applicable.

**Informed Consent Statement:** Not applicable.

**Data Availability Statement:** Not applicable.

**Acknowledgments:** We thank the three reviewers and editors for their constructive comments.

**Conflicts of Interest:** The authors declare no conflict of interest.

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
