# Peer review of "Urban Expansion Simulation Coupled with Residential Location Selection and Land Acquisition Bargaining: A Case Study of Wuhan Urban Development Zone, Central China’s Hubei Province"

_sustainability, doi:10.3390/su15010290_

Round 1

Reviewer 1 Report

The topic of this paper was important for urban land planning. The structure of the paper was clear, and the results were interesting.

I suggested the authors summarize the novelty of the paper in the Introduction section. The methods seemed to be used by the former research. Although the authors listed some problems of existing methods (line 92 and line 115), these problems seemed to be solved with some improvement. If the novelty is the coupled model, the author should further demonstrate it.

On line 35, “Less developed regions are urbanizing faster than developed regions [2]” is not absolutely accurate. This sentence may be wrong under different developing stages.

On line 37: during rapid urbanization land tends to expand in a more “aggressive” way that is faster than population growth. The sentence has grammar problems. What is “during rapid urbanization land”?

On line 59, “[14],” should be “[14].”

On line 60: Meanwhile, in a CA model, the spatial location of the cell cannot be moved so more emphasis is placed on the interactions between spatial elements and the surrounding environment…This sentence is not clear.

On line 133: it should be “…[43],…[44],…”

In figure 4, only figure 4.p had a blue color which was not explained in the legend. What did it mean and was the blue part necessary?

In figure 7, the black rectangle of a, b, c in the left is not clear.

In figure 9, I suggested the author add an explanation of type 1 and type 2, which is difficult to understand without the main context. 

Author Response

Q1: I suggested the authors summarize the novelty of the paper in the Introduction section. The methods seemed to be used by the former research. Although the authors listed some problems of existing methods (line 92 and line 115), these problems seemed to be solved with some improvement. If the novelty is the coupled model, the author should further demonstrate it.

A1: Thank you very much for your advice. We added a sentence before the last paragraph of the introduction. “However, few studies have coupled multiple decision processes to simulate urban land expansion, and decision definition needs more intelligent methods.” The three features we listed in this paragraph are the highlights of this paper.

The coupling model was compared with the traditional LRCA model, and the overall accuracy and Kappa coefficient of the coupling model were significantly higher, which proved the effectiveness of the coupling model.

Q2: On line 35, “Less developed regions are urbanizing faster than developed regions [2]” is not absolutely accurate. This sentence may be wrong under different developing stages.

A2: Indeed, the sentence has been removed.

Q3: On line 37: during rapid urbanization land tends to expand in a more “aggressive” way that is faster than population growth. The sentence has grammar problems. What is “during rapid urbanization land”?

A3: Grammatical error, this sentence has been corrected. What we want to say is that in the process of rapid urbanization, the urbanization of land is faster than the urbanization of population.

Q4: On line 59, “[14],” should be “[14].”

A4: Modified.

Q5: On line 60: Meanwhile, in a CA model, the spatial location of the cell cannot be moved so more emphasis is placed on the interactions between spatial elements and the surrounding environment…This sentence is not clear.

A5: Grammatical error, this sentence has been corrected. “Meanwhile, in a CA model, the spatial location of the cell cannot be moved. It emphasizes the interaction between spatial elements and the surrounding environment and does not explain the process and causes of urban growth.”

Q6: In figure 4, only figure 4.p had a blue color which was not explained in the legend. What did it mean and was the blue part necessary?

A6: Blue represents water. Legend has been added to the new image.

Q7: In figure 7, the black rectangle of a, b, c in the left is not clear.

A7: The picture has been redrawn and the rectangles are easier to identify.

Q8: In figure 9, I suggested the author add an explanation of type 1 and type 2, which is difficult to understand without the main context.

A8: It has been added in the article.

Reviewer 2 Report

This paper support a interesting perspective to enlarge my understanding on urban expansion simulation coupled with residential location selection and land acquisition bargaining. Generally, the part of Methodolgy is well-structured. Following are some of my questions or comments just for your consideration.

(1)   I wonder to know the way to quantify DA;

(2)   It should be more carefully to put China map (Figure 3), I suggest the author could download the official from Ministry of Natural Resources of the People’s Replublic of China, and marked clearly;

(3)   In the context of current social and economic development, especially the supply of constructed land from central government becomes more and more tightly, in other words, the tendency caculated from model would be more reliable if the paper could considering the current and future policy.

Author Response

Q1: wonder to know the way to quantify DA.

A1:DA is described in Section 2.1.2. It is difficult to quantify the benefits of DA, so we use neighborhood factors for reference [26-29].

Q2: It should be more carefully to put China map (Figure 3), I suggest the author could download the official from Ministry of Natural Resources of the People’s Replublic of China, and marked clearly;

A2: Thank you for your advice. We used data on China's administrative boundaries from the National Catalogue Service For Geographic Information (https://www.webmap.cn/commres.do?method=result100W), which is officially published by the Chinese government.

Q3: In the context of current social and economic development, especially the supply of constructed land from central government becomes more and more tightly, in other words, the tendency caculated from model would be more reliable if the paper could considering the current and future policy.

A3: Couldn't agree more. In previous studies, i.e., Reference [36], we collected land expropriation data released by relevant governments, and the results verified the validity of the proposed land expropriation bargaining model. In order to protect the interests of farmers, the model participants are assumed to be rational. The land expropriation income of each party is also calculated according to the data published by the local government, which has a certain reference.

Reviewer 3 Report

Modellin the city sprawl is a traditional and interesting topic in the field of urban geography, land use since and urban planning. This study tried to establish an urban expansion simulation model coupled with a residential location selection model and a land acquisition bargaining model. This is undoubtedly valuable. Comparing with the traditional CA model, the simulation model in this manuscript is advanced in a way. The article is clear in thought and reasonable in structure. I agree to accept this paper. It can be published after minor revisions.

    (1) The novelty of the article needs to be emphasized in the first section.

(2) Beside CA and ABM, there are some models using to simulate urban expansion, the authors should not ignore them in the introuduction.

(3) The contributions of this manscript should be emphasized in the section of conclusion.

(4) In fact, the firms especical industry firms are also important agent driving the urban sprawl. But the authors did not consider that, and why?

Author Response

Q1 The novelty of the article needs to be emphasized in the first section.

A1: Thank you very much for your advice. We added a sentence before the last paragraph of the introduction. “However, few studies have coupled multiple decision processes to simulate urban land expansion, and decision definition needs more intelligent methods.” The three features we listed in this paragraph are the highlights of this paper.

Q2: Beside CA and ABM, there are some models using to simulate urban expansion, the authors should not ignore them in the introuduction.

A2: That's true. We've added to it. “Urban expansion simulation models can be divided into top-down and bottom-up models. The top-down model mainly uses mathematical statistics or empirical equations to express the dynamic time changes and ignores the spatial-temporal differences of urban expansion, which fails to reflect the complexity of the urban system evolution process, such as the system dynamics model.”

Q3: The contributions of this manscript should be emphasized in the section of conclusion.

A3: We added a paragraph. “In this study, two crucial decision-making processes affecting urban land expansion in China are coupled, and the urban evolution of WHUDZ is simulated to reflect the impact of micro-individual decisions on global urban growth. In particular, quantifying the land expropriation process is relevant for solving land expropriation conflicts and formulating more reasonable land expropriation policies.”

Q4: In fact, the firms especical industry firms are also important agent driving the urban sprawl. But the authors did not consider that, and why?

A4: That's true. However, this study focuses on the site selection process of urban residents, assuming that all the newly added urban land is residential land. Therefore, we selected 14 factors that affect residents' location selection to conduct ANN training, and the results were satisfactory. In future studies, we need to consider the impact of more agents, such as the industry you mentioned.